# Effects of Combinations of Toxin Binders with or without Natural Components on Broiler Breeders Exposed to Ochratoxin A

**DOI:** 10.3390/ani13142266

**Published:** 2023-07-11

**Authors:** Jihwan Lee, Hyunah Cho, Dongcheol Song, Seyeon Chang, Jaewoo An, Jeonghun Nam, Byoungkon Lee, Sowoong Kim, Woo Kyun Kim, Jinho Cho

**Affiliations:** 1Department of Poultry Science, University of Georgia, Athens, GA 30602, USA; jl26112@uga.edu; 2Department of Animal Science, Chungbuk National University, Cheongju 28644, Republic of Korea; hannah0928@naver.com (H.C.); paul741@hanmail.net (D.S.); angella2425@naver.com (S.C.); blueswing547@naver.com (J.A.); 3Cherrybro Co., Ltd., Jincheon 27820, Republic of Korea; nam0353@cau.ac.kr (J.N.); bklee0418@cherrybro.com (B.L.); 4Provimi Co., Ltd., Seoul 06158, Republic of Korea; bill_so-woong_kim@provimi.com

**Keywords:** broiler breeder, ochratoxin A, toxin binder, clay mineral, natural component

## Abstract

**Simple Summary:**

Ochratoxin A (OTA) is known to be a highly toxic mycotoxin, and it is a secondary metabolite which is primarily produced by several fungi, including *Aspergillus* and *Penicillium genera*. In the poultry industry, many researchers have been searching for alternative ways to control the toxic effects of mycotoxins to safeguard the health and growth performance of birds. The clay minerals and natural components used in this study are considered useful for the mitigation of the negative effects of mycotoxins. However, only a limited number of studies have investigated the effects of supplementation with these feed additives on broiler breeders exposed to OTA. Our results indicate that toxin binders, with or without natural components, can be effective tools in the mitigation of OTA-induced problems due to their synergistic effects.

**Abstract:**

The objective of this study was to investigate the effects of toxin binders on broiler breeders fed an ochratoxin A (OTA)-contaminated diet. A total of 60 45-week-old female Arbor Acres broiler breeder birds with an initial body weight of 3.65 ± 0.35 kg were randomly divided into 6 treatment groups, with 10 replicates per group and 1 bird per replicate. The trial was conducted for 9 weeks (including 1 week of adaptation). Feed additive 1 (FA1) was composed of clay minerals (85% bentonite and 12% clinoptilolite) with 3% charcoal. FA2 was composed of clay minerals (66.1% aluminosilicates) with natural components (0.8% artichoke and rosemary plant extracts), 7% yeast extract, 0.5% beta-glucans, and 25.6% carriers. The dietary treatment groups were as follows: (1) birds fed an OTA-free basal diet (Negative Control; NC); (2) lipopolysaccharide (LPS)-challenged birds fed a diet including OTA (4 mg/kg) (Positive Control, PC); (3) the PC with 0.05% FA1 (Treatment 1, T1); (4) the PC with 0.10% FA1 (Treatment 2, T2); (5) the PC with 0.10% FA2 (Treatment 3, T3); and (6) the PC with 0.20% FA2 (Treatment 4, T4). The LPS challenge (an intramuscular injection of 1 mg *E. coli* O55:B5 LPS per kg of body weight) was performed on the first day of the experiment. The results of this experiment show that the PC treatment negatively affected (*p* < 0.05) egg production, hatchability, Haugh unit, bone mineralization, relative organ weight (abdominal fat, liver), the levels of glutamic oxaloacetic transaminase (GOT), high-density lipoprotein (HDL), and total cholesterol in the blood, and OTA accumulation in the liver compared with the NC. However, supplementation with toxin binders mitigated (*p* < 0.05) the negative effects of the OTA. Specifically, supplementation with 0.10% FA1 and 0.10% FA2 increased (*p* < 0.05) eggshell strength by week 4, and the Haugh unit and bone mineralization (phosphorous) by week 8, while decreasing (*p* < 0.05) the relative weight of the liver and the levels of GOT and HDL in the blood. Supplementation with 0.10% FA2 led to greater improvements in various parameters, including laying performance and bone mineralization, than the other treatments. In conclusion, toxin binders with or without natural components can be effective tools in the mitigation of OTA-induced problems due to their synergistic effects.

## 1. Introduction

Mycotoxins are well-known toxic secondary metabolites produced by certain fungi that grow on feed ingredients. Among them, ochratoxin A (OTA) is considered a highly toxic mycotoxin, and it is primarily produced by several fungi, including the Aspergillus and Penicillium genera [1,2]. OTA, which is the most common mycotoxin contaminant in poultry diets, is not destroyed by temperatures below 180 °C [3]. Moreover, OTA is chemically stable, so even if the fungi which produce OTA are killed, the residual toxins can still produce adverse effects, such as the contamination of an animal’s tissues, milk, and eggs, and thereby threaten human consumers [4]. Furthermore, OTA induces nephron toxicity, hepatotoxicity, teratogenicity, and immune toxicity in poultry, which are more sensitive to this toxin than mammals [5,6,7,8]. In addition to these direct and harmful effects on poultry, OTA exposure can result in the presence of OTA in the tissues and eggs of poultry. In previous studies, OTA was detected in 35% of egg samples and 41% of meat samples, and the continuous consumption of OTA was found to further increase the amount of OTA residue in eggs, and this could potentially be harmful to human health [9,10]. The supplementation of natural substances which bind mycotoxins in contaminated feed can mitigate mycotoxin-induced adverse effects. This is currently one of the most promising approaches to solving the problems outlined above [11]. Among these natural substances, clay minerals are known to have the ability to adsorb mycotoxins [12]. They also convert mycotoxins into less toxic metabolites, prevent their absorption in the intestines, and help facilitate the direct excretion of toxins in the feces [13]. Charcoal is also considered an effective toxin binder. Numerous in vitro studies have reported that charcoal can effectively absorb mycotoxins and alleviate mycotoxin-induced immune suppression, consequently improving the growth performance of broilers in vivo [14,15]. In addition to the use of these toxin binders, several additional strategies have been proposed to minimize the harmful effects of mycotoxins [16]. Interest in the effects of plant extracts, yeast extracts, and prebiotics on mycotoxins has recently increased because of their ability to minimize fungal and bacteria growth [17,18,19,20]. Moreover, lipopolysaccharide (LPS) has been widely used to induce immune stress in animal models [21]. Many studies have shown that LPS-induced immune stress can cause various negative effects in poultry, such as reduced growth performance, intestinal mucosal damage, and immunosuppression [22]. In rodents, oral challenges with LPS disturbed the subjects’ response to other xenobiotic agents and thus increased the toxicity of trichothecenes such as deoxynivalenol [23]. Therefore, we simultaneously conducted an OTA challenge via feed and intramuscular LPS injections to maximize the negative effects of OTA in broiler breeders. However, only a limited number of studies have investigated the synergistic effects achieved by using other natural substances (i.e., plant extracts, yeast extracts and prebiotics) along with toxin binders (i.e., clay minerals and charcoals). Therefore, this study was conducted to investigate the effects of toxin binders (i.e., clay minerals and charcoals), with or without natural substances (i.e., plant extracts, yeast extracts, and prebiotics), on the laying performance, egg quality, relative organ weight, bone mineralization, blood traits, and toxin concentrations of broiler breeders challenged with ochratoxin A and LPS.

## 2. Materials and Methods

### 2.1. Ethics

The experimental protocols describing the management and care of animals were reviewed and approved by the Animal Care and Use Committee of Chungbuk National University (Cheongju, Republic of Korea; CBNUA-2045-22-01).

### 2.2. Preparation of Ochratoxin A

OTA was cultured according to the method of Ruan et al. [24]. In moldy corn, *Aspergillus ochraceus* was identified using Czapek Dox Agar (MB cell, Seoul, Republic of Korea), and then was enriched to 8 × 10^10^ CFU/mL using Czapek Dox Media (CDM) (MB cell, South Korea). Corn was inoculated with this *A. ochraceus* suspension twice a day (08:00 and 20:00) and cultured at 29 °C. On day 14 of incubation, contaminated corn was terminated by autoclaving at 121 °C for 15 min. OTA content was measured using ELISA Kits (Romer AgraQuant, Romer Labs, Jalan Bukit Merah, Singapore).

### 2.3. Experimental Animal and Design

A total of sixty 45-week-old Arbor Acres female broiler breeders (initial body weight of 3.65 ± 0.35 kg) were used in this experiment. All birds were randomly allocated to six dietary treatments in a randomized complete block design. The experiment was conducted for 9 weeks including a 1 week adaptation period. Feed additive 1 (FA1) was composed of 85% clinoptinolite, 12% bentonite and 3% charcoal. FA2 was composed of 66.1% aluminosilicates, 0.8% plant extracts (artichoke and rosemary), 7% yeast extracts, 0.5% beta-glucans and 25.6% carriers. The dietary treatments were as follows: (1) Negative Control (NC): OTA free diets; (2) Positive Control (PC): OTA (4 mg/kg) with LPS challenge; (3) Treatment 1 (T1): PC with 0.05% FA1 in the diet; (4) Treatment 2 (T2): PC with 0.10% FA1; (5) Treatment 3 (T3): PC with 0.10% FA2; and (6) Treatment 4 (T4): PC with 0.20% FA2. LPS challenge was performed on the first day of the experiment, and 1 mg of *E. coli* O55:B5 LPS per kg of body weight was intramuscularly injected according to the method of Xu et al. [25]. Experimental feed was provided by Cherry Buro Co., Ltd. (Jincheon, Republic of Korea; Table 1). The broiler breeders were raised in steel cages (W: 59 cm, D: 43 cm, H: 51 cm). All birds were given limited access to feed and water throughout the experiment. During the experiment, all chickens were housed in a room with 16 h of light and 8 h of darkness.

### 2.4. Laying Performance

During the experiment period, eggs were collected at 3:00 p.m. every day. The egg production values were calculated for the total number of eggs (normal eggs, broken eggs, and shell less eggs) and for the number of normal eggs specifically. Egg weight was calculated by dividing the total normal egg weight by the total number of normal eggs.

### 2.5. Fertility and Hatchability

During the entire period, broken, contaminated, deformed, and small eggs (under 48 g) were excluded from fertility and hatchability evaluations. Fertility was calculated on the 8th day of the incubation by dividing the number of fertilized eggs by the total number of eggs incubated. Hatchability was calculated by dividing the number of new hatched chicks by the number of eggs in the incubator (MX-1000CD, Gimhae, Republic of Korea).

### 2.6. Egg Quality

Eggshell strength was measured using a texture analyzer (model 081002, FHK, Fujihara Ltd., Tokyo, Japan) and in units of compressive force loaded onto a unit of eggshell surface area. Eggshell thickness was measured at three random locations (top, middle, and bottom) using a micrometer dial pipe gauge (model 7360, Mitutoyo Co., Ltd., Kawasaki, Japan), and the average value was used to calculate the eggshell strength. Egg shell color was measured at the bottom of the egg using a colorimeter (model CM-25cG, konica minolta, Osaka, Japan) for L* (lightness), a* (redness), and b* (yellowness). Egg yolk color was measured using a colorimetric Roche color fan (Hoffman-La Roche Ltd., Basel, Switzerland).
1 (*pale yellow*)~15 (*deep orange*)

The Haugh unit (HU) was measured using a tripod micrometer (AMES, Waltham, MA, USA) for albumin height after breaking the egg on a flat glass surface, and the HU value was calculated based on the equation below, where H represents albumin length (mm) and W represents egg weight (g).
HU *=* 100 log (H + 7.57 − 1.7 W^0.37^)

### 2.7. Bone Mineralization

At the end of the experiment, the right tibiae of the slaughtered broiler breeders were removed for bone mineralization evaluation. To establish the ash content, the constant weight of the crucible was obtained using the direct incineration method. An approximately 1 g bone sample was taken, incinerated in a furnace (Ward, South Australia, Australia) at 200 °C for 1 h, 300 °C for 1 h, and 600 °C for 3 h, and then cooled in a desiccator for 30 min. The ash sample was cooled and weighed, and the amount of ash was calculated from the difference in weight. The ash samples were dissolved in a hydrochloric acid solution (1:1, *v*/*v*). Calcium (Ca) levels were measured with an atomic absorption spectrophotometer (AAS-3300, PERKIN ELMER, Norwalk, CT, USA), and phosphorus (P) levels were measured at 470 nm with a spectrophotometer (UV-2450, Shimadzu Co., Tokyo, Japan) according to the molybdenum blue colorimetric method [26].

### 2.8. Relative Organ Weight

At the end of the experiment, the weights of the liver, spleen and abdominal fat were measured. These values were expressed in terms of relative weight to live weight.
Relative organ weight (g/kg) = organ weight (g)/body weight (kg)

### 2.9. Blood Profile

At the end of the experiment, blood samples were collected with tubes (Becton Dickinson, Franklin Lakes, NJ, USA) from the wing vein in all birds. The serum was separated by means of centrifugation at 3000 rpm and 4 °C for 20 min. High-density lipoprotein (HDL) and total cholesterol (Total-C) concentrations in the blood were analyzed using an automatic blood analyzer (ADVIA 1650, Bayer, Tokyo, Japan). Glutamic oxaloacetic transaminase (GOT) and glutamic pyruvic transaminase (GPT) activities were analyzed using an automatic biochemistry analyzer (Hitachi 747, Hitachi, Tokyo, Japan).

### 2.10. OTA Accumulation in the Feed and Liver

After the experiment was completed, the feed samples were collected before starting the experiment and stored at −20 °C until analysis. Liver samples from all individual birds were collected and weighted. The samples were ground using liquid nitrogen and stored at −20 °C until analysis. OTA analysis was performed by extracting 2.5 g of the liver with 20 mL of 1% NaHCO3/methanol (30:70, *v*/*v*). After centrifugation at 3000 rpm for 10 min, 12 mL of the supernatant was mixed with an equal volume of PBS and applied to an OchraTest WB immunocolumn (Vicam, Fleurus, Belgium). The eluate was filtered through a 0.45 μL syringe filter and analyzed using high-pressure liquid chromatography (HPLC Prominence, Shimadzu, Kyoto, Japan) equipped with a fluorescence detector. HPLC separations were performed using a Phenomenex Luna C18(2) 3 μm, 150 4.60 mm column equipped with a Gemini C18, 4.3 mm Security Guard pre-column (Phenomenex Inc., Torrance, CA, USA). A flow rate of 0.7 mL/min, an injection volume of 40 L, and a column oven temperature of 30 °C were set. Fluorescence detection was performed at an excitation wavelength of 327 nm and an emission wavelength of 462 nm. Conformity with the retention time of the standard product measured at the same time was the criterion for the qualitative test. A calibration curve based on the analytical concentration of the standard product was generated, and the obtained peak height was used for the quantitative test.

### 2.11. Statistical Analysis

All data were analyzed by means of one-way ANOVA using SPSS software (ver. 25.0; IBM, Armonk, NY, USA). Tukey’s multiple range test examined differences among treatment groups, which were considered significant at *p* < 0.05.

## 3. Results

### 3.1. Laying Performance

The OTA-contaminated diet (PC) significantly decreased (*p* < 0.05) egg production compared to the negative control (NC) (Table 2). However, supplementation with 0.10% FA2 in the OTA-contaminated diet (T3) significantly increased egg production compared to the PC. There were no significant differences in egg production among the natural-additive-supplemented groups (T1, T2, T3 and T4).

### 3.2. Fertility and Hatchability

The OTA-contaminated diet (PC) significantly decreased (*p* < 0.05) hatchability compared to the NC group (Table 3). Although there was no significant difference (*p* > 0.05), the natural-feed-additive-supplemented groups showed numerically increased hatchability compared to the PC group.

### 3.3. Egg Quality

There was no significant difference (*p* > 0.05) in egg quality at 4 and 8 weeks, except for the Haugh unit (HU) and eggshell strength, among the treatments (Table 4 and Table 5). The OTA-contaminated diet (PC) significantly decreased (*p* < 0.05) HU at 4 and 8 weeks compared to the NC group. At 4 weeks, supplementation with FA1 and FA2 increased HU to similar levels to the NC group. At 8 weeks, supplementation with 0.10% FA1 (T2) and 0.10% FA2 (T3) significantly improved HU compared to the PC group. Regarding eggshell strength, the PC demonstrated significantly reduced eggshell strength compared to the NC group at 4 weeks, whereas supplementation with 0.10% FA1 (T2) and 0.10% FA2 (T3) significantly increased eggshell strength compared to the PC group.

### 3.4. Bone Mineralization

The OTA-contaminated diet (PC) significantly decreased (*p* < 0.05) ash, Ca and *p* content in the tibia compared to the NC group (Table 6). Supplementation with 0.10% FA2 (T3) significantly increased (*p* < 0.05) ash content in the tibia in comparison to the PC group. The FA2-supplemented groups (T3 and T4) had a higher Ca content in the tibia than the PC group. Moreover, supplementation with 0.10% FA1 (T2) and two levels of FA2 significantly increased P content in the tibia compared to the PC group.

### 3.5. Relative Organ Weight

The OTA-contaminated diet (PC) significantly increased (*p* < 0.05) the relative weight of the liver and abdominal fat in comparison to the NC group (Table 7). However, all of the natural-feed-additive-supplemented groups (T1–T4) displayed a lower relative liver weight than the PC group. In particular, supplementation with 0.10% FA1 (T2) and 0.10% FA2 (T3) significantly decreased (*p* < 0.05) the relative weight of the liver to similar values to the NC group. Supplementation with 0.10% FA2 (T3) alleviated (*p* < 0.05) the increased relative weight of abdominal fat caused by the OTA-contaminated diet (PC) and even decreased (*p* < 0.05) it to similar values to the NC group. There was no significant difference (*p* > 0.05) in the relative weight of the spleen among the treatment groups.

### 3.6. Blood Profiles

The OTA-contaminated diet (PC) significantly increased (*p* < 0.05) GOT and HDL levels in blood, whereas it decreased (*p* < 0.05) total cholesterol levels in the blood in comparison to the NC group (Table 8). In particular, supplementation with two levels of FA1 and 0.10% FA2 significantly mitigated (*p* < 0.05) the increased GOT levels in the blood caused by the OTA-contaminated diet (PC). The natural-feed-additive-supplemented groups demonstrated significantly increased (*p* < 0.05) total cholesterol levels in the blood, displaying similar values to the NC group. Supplementation with 0.10% FA1 and two levels of FA2 remarkably decreased (*p* < 0.05) HDL levels in the blood to similar values to the NC group. There was no significant difference in the GPT levels in the blood among the treatment groups.

### 3.7. OTA Accumulation

The OTA-contaminated diet (PC) significantly increased (*p* < 0.05) OTA accumulation in the feed and liver in comparison to the NC group (Table 9). Supplementation with natural feed additives significantly decreased (*p* < 0.05) OTA accumulation in the liver compared to the PC group.

## 4. Discussion

In the current study, ochratoxin A (OTA)-contaminated diets caused significantly lower egg production and hatchability and numerically decreased egg weight compared to the negative control. This result is in agreement with the studies of Zahoor-ul-Hassan et al. [9] and Eid et al. [27], who reported that ochratoxin caused lower egg mass, egg production, average egg weight, and hatchability along with reduced feed consumption. According to a previous study, OTA has the highest embryotoxicity among all mycotoxins, and thus can increase embryo mortality and decrease fertilized egg hatchability [5]. Moreover, Konrad and Röschenthaler [28] reported that OTA can negatively influence the synthesis of deoxyribonucleic acid (DNA), ribonucleic acid (RNA), and proteins. These authors suggested that the reduction in egg performance may be attributed to these disorders in metabolic pathways caused by OTA.

The supplementation of FA1, which mainly consisted of toxin binders (i.e., clay mineral and charcoal), and FA2, which consisted of toxin binders with natural components (i.e., clay mineral, plant extract, yeast extract and beta-glucan), used in this study mitigated the reduced egg production and hatchability caused by OTA, respectively. Toxin binders such as clay mineral and charcoal, which are the main components in FA1, bind and immobilize the mycotoxin in the gastrointestinal tract (GIT), thereby reducing the absorption of toxins into the GIT in poultry [29,30,31].

In this study, the improved egg production and hatchability caused by supplementing toxin binders (i.e., clay mineral and charcoal) may be attributed to the abovementioned mechanism and the beneficial roles of binders in minimizing the harmful effects of mycotoxins. Likewise, the addition of natural feed additives (i.e., plant extract, yeast extract and beta-glucan) improved egg production and hatchability; the group supplemented with 0.10% natural additives (T3) had the highest egg production and hatchability among the treatments. Thanissery et al. [32] and Esper et al. [33] reported that plant and yeast extracts are effective against molds such as *Aspergillus* and *A. fumigatus* and bacteria such as *Clostridium perfringens*. Moreover, previous studies suggested that plant extracts and prebiotics stimulated the release of endogenous digestive enzymes, which enhance nutrient digestion [34,35]. As mentioned above, mycotoxins can interfere with metabolic pathways such as carbohydrate, protein and lipid metabolisms. However, natural components such as plant extracts, yeast extracts, and beta-glucan could support these metabolisms by secreting endogenous enzymes and inhibiting the growth of molds as a fundamental cause. Thus, the improved egg production and hatchability observed in the current study may be due to the beneficial effects of toxin binders and natural components. Moreover, the Haugh units at 8 weeks were enhanced by supplementing FA1 and FA2 in the current study. Our results regarding Haugh unit improvement provide clear evidence supporting the hypothesis that the addition of natural feed additives can improve digestibility by enhancing enzyme activity.

It has been reported that egg quality, bone mineralization, and the relative weight of the liver and abdominal fat can be negatively affected by mycotoxins. Egg internal and shell quality are important parameters for the worldwide egg industry [36]. In the current study, eggshell strength at 4 weeks and Haugh units at 8 weeks were decreased, while the OTA-contaminated diet increased the relative weight of the liver and abdominal fat. Jia et al. [36] observed the lowest eggshell strength in the mycotoxin-affected groups. According to Zaghini et al. [37], eggshell strength is affected by shell thickness, and they observed reduced eggshell thickness along with eggshell strength. This is consistent with the current study which indicated that shell thickness in the OTA-contaminated group (PC) was noticeably thinner than in the negative control. This poor eggshell quality may be due to poor calcium and phosphorus absorption, supported by bone mineralization in the current study. We found that the OTA-contaminated diet caused a reduction in ash, Ca and P contents in the tibia. Yildirim et al. [38] reported that high levels of mycotoxin in diets can reduce the circulation level of parathyroid hormone (PTH), which is an essential regulator of extracellular calcium and phosphate metabolism. PTH increases Ca re-absorption and P excretion in the kidney and enhances Ca absorption in the intestine by stimulating the synthesis of 1,25-dihydroxy vitamin D in the kidney [39]. The dysregulation of the above-mentioned mechanisms may negatively affect egg quality and bone mineralization.

In the current study, the addition of FA1 and FA2 alleviated poor eggshell strength at 4 weeks and bone mineralization in the tibia. These results are in agreement with the results of Darmawan and Ozturk [40] showing that eggshell strength linearly increased with increasing clay mineral levels. According to the study by Elliott et al. [41], clay minerals have a high mineral content, ion exchange capacity, and calcium affinity; these functions positively affect eggshell quality. Kriseldi et al. [42] suggested that clay minerals increased alkaline phosphatase, influencing bone mineralization and phytate degradation in the small intestine.

Plant extracts, yeast extracts, and prebiotics have been reported to affect eggshell formation. Lokaewmanee et al. [43] reported that plant extracts such as red clover and garlic enhanced eggshell strength, and this beneficial result could be attributed to the improvement in the morphology of the small intestine. Moreover, according to a review by Miranda et al. [44], plant extracts mitigated pro-inflammatory response and improved antioxidant enzyme release, thereby increasing bone repair, osteogenesis, formation and mineralization of bone. Thus, their beneficial functions may cause the improvement in Ca and P availability for eggshell formation. However, numerous studies reported that yeast and prebiotics did not affect eggshell formation [45,46]. The current study indicated that improved eggshell and bone mineralization may be due to clay minerals and plant extracts.

In the current study, the OTA-contaminated diet increased the relative weight of the liver and abdominal fat while decreasing total cholesterols in the blood compared to the negative control. Our results are in agreement with previous studies showing that mycotoxins, including deoxynivalenol (DON), zearalenone (ZEA), and OTA, caused fat deposition in the liver and reductions in cholesterol and triglyceride contents in the blood [37,47,48,49]. These authors suggested that triglyceride and cholesterol in chickens fed mycotoxins-contaminated diets were transported into the liver, and thus fat accumulation in the liver was increased, while cholesterol and triglyceride levels in the blood were decreased. Our results agree with the abovementioned mechanisms. Furthermore, in the current study, we found that the OTA-contaminated diet caused high OTA accumulation in the liver and GOT concentration in the blood. These findings are in agreement with previous studies reporting that chickens exposed to mycotoxins had high mycotoxin accumulation in the liver and activated GOT in the serum [11,50,51]. Mycotoxins are mainly metabolized in the liver and transformed into their reactive metabolites, and thus cause hepatocyte cancerization and liver damage [52]. According to Rashidi et al. [49], mycotoxins induced increased hepatocyte permeability, and then transaminases could shift from the damaged hepatocyte to the bloodstream, consequently causing increased activity of transaminases like GOT and GPT. The increased levels of GOT in the blood and OTA accumulation in the liver observed in the current study may be explained by the abovementioned mechanisms. Clay minerals have been known to have high mycotoxin adsorption capacity due to their high surface area and ion exchange capacity [17,53]. Moreover, plant extracts that have antimicrobial and antifungal properties can inhibit bacteria and mold [54]. In the current study, supplementation with natural feed additives (FA1 and FA2) mitigated the relative weight of the liver and abdominal fat, the GOT level in the blood, and OTA accumulation in the liver. These findings may be attributed to the inhibition of OTA by clay minerals and plant extracts.

## 5. Conclusions

In the current study, we found that supplementation with toxin binders (FA1) and toxin binders with natural components (FA2) mitigated the negative effects of OTA and even alleviated Haugh units, the relative weight of the liver and abdominal fat, total cholesterol, and HDL, bringing them to similar levels to those observed in non-challenged treatments (NC group). In particular, the group supplemented with 0.10% FA2 demonstrated improved parameters compared to the other treatment groups. The results of our study indicate that toxin binders with or without natural components can be very useful in the mitigation of OTA-contamination-induced problems by inhibiting OTA and improving nutrient absorption and host health. Moreover, we observed tenuous synergistic effects between clay minerals and natural components.

## Figures and Tables

**Table 1 animals-13-02266-t001:** Ingredients and chemical composition of the basal experimental diets (as fed basis).

Items	0–8 Weeks
Ingredients (%)	
Corn	54.12
Soybean meal, 45%	15.00
DDGS 28%	10.00
Corn Gluten Feed	6.27
Wheat Pollards	2.50
Rice pollards	1.50
Animal Fats	0.50
L-Lys-SO_4_	0.08
DL-Methionine	0.12
L-Threonine	0.02
L-Tryptophan	0.13
Salt	0.21
Limestone	8.97
Mineral premix ^1^	0.22
Vitamin premix ^2^	0.11
Choline	0.25
Total	100
Calculated Value	
ME, Kcal/Kg	2710
Crude protein, %	15.42
Crude Fat, %	3.86
Crude Fiber, %	3.85
Total Lys, %	0.749
Total TSAA, %	0.684
Calcium, %	3.50
Available P, %	0.41

^1^ Provided per kg of diet: 37.5 mg Zn (as ZnSO_4_), 37.5 mg of Mn (MnO_2_), 37.5 mg of Fe (as FeSO_4_•7H_2_O), 3.75 mg of Cu (as CuSO_4_•5H_2_O), 0.83 mg of I (as KI), and 0.23 mg of Se (as Na_2_SeO_3_•5H2O). ^2^ Provided per kg of diet: 15,000 IU of vitamin A, 3750 IU of vitamin D3, 37.5 mg of vitamin E, 2.55 mg of vitamin K3, 3 mg of thiamin, 7.5 mg of riboflavin, 4.5 mg of vitamin B6, 24 μg of vitamin B12, 51 mg of niacin, 1.5 mg of folic acid, 0.2 mg of biotin and 13.5 mg of pantothenic acid.

**Table 2 animals-13-02266-t002:** Effect of combinations of natural feed additives on egg production and egg weight in broiler breeders fed with an ochratoxin A diet ^1^.

Items	NC	PC	T1	T2	T3	T4	SEM	*p*-Value
Egg production, %	95.00 a	79.91 c	83.13 bc	86.11 bc	88.49 b	82.54 bc	1.763	<0.001
Egg weight, g/egg	64.59	62.58	63.40	64.86	63.74	63.68	0.761	0.401

^1^ Abbreviations: NC = Negative Control, basal diet; PC = Positive Control, basal diet with + 4 ppm ochratoxin A (OTA) + lipopolysaccharide (LPS) challenge; T1 = basal diet with FA1 including 0.05% clay mineral and charcoal + 4 ppm OTA + LPS challenge; T2 = basal diet with FA1 including 0.10% clay mineral and charcoal + 4 ppm OTA + LPS challenge; T3 = basal diet with FA2 including 0.10% clay mineral, plant extract, yeast extract and beta-glucan + 4 ppm OTA + LPS challenge; T4 = basal diet with FA2 including 0.20% clay mineral, plant extract, yeast extract and beta-glucan + 4 ppm OTA + LPS challenge; SEM, standard error of mean. a–c = Means with different letters are significantly different (*p* < 0.05). N = 10.

**Table 3 animals-13-02266-t003:** Effect of combinations of natural feed additives on fertility and hatchability in broiler breeders fed with an ochratoxin A diet ^1^.

Items	NC	PC	T1	T2	T3	T4	SEM	*p*-Value
Fertility, %	91.67	85.94	85.65	87.96	86.11	86.57	1.730	0.119
Hatchability, %	86.25 a	75.26 b	77.08 ab	78.24 ab	78.70 ab	77.31 ab	2.282	0.024

^1^ Abbreviations: NC = Negative Control, basal diet; PC = Positive Control, basal diet with + 4 ppm ochratoxin A (OTA) + lipopolysaccharide (LPS) challenge; T1 = basal diet with FA1 including 0.05% clay mineral and charcoal + 4 ppm OTA + LPS challenge; T2 = basal diet with FA1 including 0.10% clay mineral and charcoal + 4 ppm OTA + LPS challenge; T3 = basal diet with FA2 including 0.10% clay mineral, plant extract, yeast extract and beta-glucan + 4 ppm OTA + LPS challenge; T4 = basal diet with FA2 including 0.20% clay mineral, plant extract, yeast extract and beta-glucan + 4 ppm OTA + LPS challenge; SEM, standard error of mean. a–b = Means with different letters are significantly different (*p* < 0.05). N = 10.

**Table 4 animals-13-02266-t004:** Effect of combinations of natural feed additives on egg quality in broiler breeders fed with an ochratoxin A diet after 4 weeks of the experiment ^1^.

Items	NC	PC	T1	T2	T3	T4	SEM	*p*-Value
Eggshell color (Hunter color)								
L*	72.39	77.37	77.47	80.73	78.07	78.49	1.769	0.091
a*	8.11	6.98	7.67	6.10	8.59	7.48	0.692	0.377
b*	21.25	22.09	21.82	20.92	23.26	21.33	0.904	0.634
Egg yolk color	8.00	7.25	7.22	7.56	7.44	7.44	0.358	0.662
Haugh unit	86.24 a	78.51 b	83.28 ab	84.18 ab	80.99 ab	84.24 ab	1.561	0.030
Eggshell strength (kg/cm^2^)	4.81 a	4.18 c	4.38 bc	4.54 b	4.55 b	4.29 c	0.053	<0.001
Eggshell Thickness (μm)	411.27	404.83	404.56	407.33	408.48	405.67	6.211	0.972

^1^ Abbreviations: NC = Negative Control, basal diet; PC = Positive Control, basal diet with + 4 ppm ochratoxin A (OTA) + lipopolysaccharide (LPS) challenge; T1 = basal diet with FA1 including 0.05% clay mineral and charcoal + 4 ppm OTA + LPS challenge; T2 = basal diet with FA1 including 0.10% clay mineral and charcoal + 4 ppm OTA + LPS challenge; T3 = basal diet with FA2 including 0.10% clay mineral, plant extract, yeast extract and beta-glucan + 4 ppm OTA + LPS challenge; T4 = basal diet with FA2 including 0.20% clay mineral, plant extract, yeast extract and beta-glucan + 4 ppm OTA + LPS challenge; L*, lightness; a*, redness; b*, yellowness; SEM, standard error of mean. a–c = Means with different letters are significantly different (*p* < 0.05). N = 10.

**Table 5 animals-13-02266-t005:** Effect of combinations of natural feed additives on egg quality in broiler breeders fed with and ochratoxin A diet after 8 weeks of the experiment ^1^.

Items	NC	PC	T1	T2	T3	T4	SEM	*p*-Value
Eggshell color (Hunter color)								
L*	73.81	78.79	78.89	82.15	79.49	79.91	1.788	0.094
a*	9.53	8.40	9.09	7.51	10.01	8.90	0.692	0.382
b*	22.67	23.51	23.24	22.03	24.68	22.75	0.908	0.560
Egg yolk color	8.30	7.50	7.67	8.11	8.22	7.78	0.291	0.327
Haugh unit	87.19 a	70.69 c	78.84 abc	82.20 ab	85.95 ab	77.91 bc	2.115	<0.001
Eggshell strength (kg/cm^2^)	4.42	3.49	3.92	3.40	3.36	3.87	0.358	0.266
Eggshell Thickness (μm)	415.50	396.08	397.52	416.52	406.00	402.19	10.972	0.686

^1^ Abbreviations: NC = Negative Control, basal diet; PC = Positive Control, basal diet with + 4 ppm ochratoxin A (OTA) + lipopolysaccharide (LPS) challenge; T1 = basal diet with FA1 including 0.05% clay mineral and charcoal + 4 ppm OTA + LPS challenge; T2 = basal diet with FA1 including 0.10% clay mineral and charcoal + 4 ppm OTA + LPS challenge; T3 = basal diet with FA2 including 0.10% clay mineral, plant extract, yeast extract and beta-glucan + 4 ppm OTA + LPS challenge; T4 = basal diet with FA2 including 0.20% clay mineral, plant extract, yeast extract and beta-glucan + 4 ppm OTA + LPS challenge; L*, lightness; a*, redness; b*, yellowness; SEM, standard error of mean. a–c = Means with different letters are significantly different (*p* < 0.05). N = 10.

**Table 6 animals-13-02266-t006:** Effect of combinations of natural feed additives on bone mineralization in broiler breeders fed with an ochratoxin A diet ^1^.

Items	NC	PC	T1	T2	T3	T4	SEM	*p*-Value
Ash, %	31.75 a	22.27 c	23.25 c	25.12 bc	27.40 b	26.19 bc	0.980	<0.001
Ca, %	10.73 a	5.87 c	6.21 c	6.75 c	8.80 b	7.97 b	0.317	<0.001
P, %	5.04 a	3.29 c	3.72 bc	4.04 b	4.31 b	4.12 b	0.189	<0.001

^1^ Abbreviations: NC = Negative Control, basal diet; PC = Positive Control, basal diet with + 4 ppm ochratoxin A (OTA) + lipopolysaccharide (LPS) challenge; T1 = basal diet with FA1 including 0.05% clay mineral and charcoal + 4 ppm OTA + LPS challenge; T2 = basal diet with FA1 including 0.10% clay mineral and charcoal + 4 ppm OTA + LPS challenge; T3 = basal diet with FA2 including 0.10% clay mineral, plant extract, yeast extract and beta-glucan + 4 ppm OTA + LPS challenge; T4 = basal diet with FA2 including 0.20% clay mineral, plant extract, yeast extract and beta-glucan + 4 ppm OTA + LPS challenge; Ca, calcium; P, phosphorus; SEM, standard error of mean. a–c = Means with different letters are significantly different (*p* < 0.05). N = 10.

**Table 7 animals-13-02266-t007:** Effect of combinations of natural feed additives on the relative weights of the liver, spleen and abdominal fat in broiler breeders fed with an ochratoxin A diet ^1^.

Items	NC	PC	T1	T2	T3	T4	SEM	*p*-Value
Liver, g/kg	19.55 c	24.56 a	22.38 b	21.27 bc	20.81 bc	21.73 b	0.498	<0.001
Spleen, g/kg	0.73	0.67	0.70	0.68	0.71	0.68	0.021	0.337
Abdominal fat, g/kg	28.57 b	31.74 a	30.70 ab	29.51 ab	28.60 b	30.02 ab	0.654	0.012

^1^ Abbreviations: NC = Negative Control, basal diet; PC = Positive Control, basal diet with + 4 ppm ochratoxin A (OTA) + lipopolysaccharide (LPS) challenge; T1 = basal diet with FA1 including 0.05% clay mineral and charcoal + 4 ppm OTA + LPS challenge; T2 = basal diet with FA1 including 0.10% clay mineral and charcoal + 4 ppm OTA + LPS challenge; T3 = basal diet with FA2 including 0.10% clay mineral, plant extract, yeast extract and beta-glucan + 4 ppm OTA + LPS challenge; T4 = basal diet with FA2 including 0.20% clay mineral, plant extract, yeast extract and beta-glucan + 4 ppm OTA + LPS challenge; SEM, standard error of mean. a–c = Means with different letters are significantly different (*p* < 0.05). N = 10.

**Table 8 animals-13-02266-t008:** Effect of combinations of natural feed additives on the blood profile in broiler breeders fed with an ochratoxin A diet ^1^.

Items	NC	PC	T1	T2	T3	T4	SEM	*p*-Value
GOT, U/L	137.00 d	229.88 a	198.44 bc	172.89 c	191.56 bc	213.56 ab	5.280	<0.001
GPT, U/L	3.70	4.75	4.33	4.11	4.33	4.22	0.342	0.442
Total-C, mg/dL	119.70 a	95.25 b	119.22 a	116.89 a	119.33 a	117.44 a	4.373	0.005
HDL, mg/dL	49.00 c	78.25 a	70.11 ab	52.44 c	53.33 c	60.33 bc	3.948	<0.001

^1^ Abbreviations: NC = Negative Control, basal diet; PC = Positive Control, basal diet with + 4 ppm ochratoxin A (OTA) + lipopolysaccharide (LPS) challenge; T1 = basal diet with FA1 including 0.05% clay mineral and charcoal + 4 ppm OTA + LPS challenge; T2 = basal diet with FA1 including 0.10% clay mineral and charcoal + 4 ppm OTA + LPS challenge; T3 = basal diet with FA2 including 0.10% clay mineral, plant extract, yeast extract and beta-glucan + 4 ppm OTA + LPS challenge; T4 = basal diet with FA2 including 0.20% clay mineral, plant extract, yeast extract and beta-glucan + 4 ppm OTA + LPS challenge; GOT, glutamic oxaloacetic transaminase; GPT, glutamic pyruvic transaminase; Total-C, total cholesterol; HDL, high-density lipoprotein; SEM, standard error of mean. a–d = Means with different letters are significantly different (*p* < 0.05). N = 10.

**Table 9 animals-13-02266-t009:** Effect of combinations of natural feed additives on OTA deposition in the liver and feed of broiler breeders fed with an ochratoxin A diet ^1^.

Items	NC	PC	T1	T2	T3	T4	SEM	*p*-Value
Feed, ppm	ND	4.02	4.01	4.00	4.01	4.01	0.003	0.173
Liver, ppb	ND	48.07 a	33.18 b	30.79 bc	29.13 c	33.19 b	0.760	<0.001

^1^ Abbreviation: NC = Negative Control, basal diet; PC = Positive Control, basal diet with + 4 ppm ochratoxin A (OTA) + lipopolysaccharide (LPS) challenge; T1 = basal diet with FA1 including 0.05% clay mineral and charcoal + 4 ppm OTA + LPS challenge; T2 = basal diet with FA1 including 0.10% clay mineral and charcoal + 4 ppm OTA + LPS challenge; T3 = basal diet with FA2 including 0.10% clay mineral, plant extract, yeast extract and beta-glucan + 4 ppm OTA + LPS challenge; T4 = basal diet with FA2 including 0.20% clay mineral, plant extract, yeast extract and beta-glucan + 4 ppm OTA + LPS challenge; ND, non-detect; SEM, standard error of mean. a–c = Means with different letters are significantly different (*p* < 0.05). N = 10.

## Data Availability

No new data were generated in this manuscript.

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
