# Peer review of "Effects of Combinations of Toxin Binders with or without Natural Components on Broiler Breeders Exposed to Ochratoxin A"

_animals, 2023, doi:10.3390/ani13142266_

Round 1

Reviewer 1 Report

Dear authors,

in this paper you show the impact of toxin binder, combined with/without natural ingredients, on different parameters of poultry, after exposure to ochratoxin in food.

General comments:

Title is too long, please try to merge all tested parameters into a more appropriate shorter title

All the tables - is there a clear need to have almost the same explanations behind every table?

Material and Methods:

- please describe how was OTA added to the poultry feed, at what amount? Was contaminated corn source of OTA?

- I am not sure did I get it right, but if you mention fertilized eggs, it means that you had some roosters in the experimental flock? You are mentioning only female broiler breeders?

Specific comments:

L 16 - penicillium with capital letter (as in L 53)

L 34-35 birds received up to 1ml intramuscularly? as in 2.3.; it was in total around 4ml?

L 43 sentence needs revision (is numerically improved parameters?)

L 79 no need for - behind LPS

L 98, 101 ittalic Aspergillus

Last sentence in 2.3. - "it was fixed to..." needs revision

L 111 - shell less?

L 115-116 - ...small eggs under 48g eggs?

L 117 - what is a breeding room?

L 134 - here mentioned formula is related to?

L 144- slaughtered broiler breeders - males? females?

L 150 ahsed?

L 157 - how was the live weight of the bird determined throughout the experiment?

L 164 -which subdermal vein?

L 163-169 please mention first full name and then abbreviation in the brackets

L 173 - how were the samples grounded?

L 195, 211,225,236,249,262 - please change the sentences "By supplementation")

Tables 2-5, what does N=10 means?

L 262 - OTA in the feed? accumulation not mentioned before in the Material and Methods?

L 277 - dot

L 311-312 sentence needs revision

L 344 cholestrerol?

minor changes needed

Author Response

Reviewer #1:

1. Title is too long, please try to merge all tested parameters into a more appropriate shorter title.

Answer: Thank you for pointing this out. I have revised title from “Effects of combination of toxin binder with or without natural components on laying performance, egg quality, bone mineralization, relative organ weight, blood profile, ochratoxin A accumulation of liver in broiler breeders exposed to ochratoxin A” to “Effects of combination of toxin binder with or without natural components in broiler breeders exposed to ochratoxin A

  1. Is there a clear need to have almost the same explanation behind every table?

Answer: Thank you for pointing this out. Because all description in table is the information about treatment (NC to T4), a common description is included. The description after that is slightly different because they are information of abbreviations for the parameters.

  1. Please describe how was OTA added to the poultry feed, at what amount? Was contaminated corn source of OTA?

Answer: Thank you for pointing this out. I have showed the method how to add OTA to feed (Line 97-104). As you said, contaminated corn was main source of OTA.

We collected Aspergillus ochraceus using Czapek Dox Agar from moldy corn and then enriched to 8 × 1010 CFU/mL using Czapek Dox Media (CDM) (MB cell, South Korea). And, Corn was inoculated with A. ochraceus suspension twice a day (08:00 and 20:00) and cultured at 29 °C. On day 14 of incubation, contaminated corn was terminated by autoclaving at 121 °C for 15 min. When we measured OTA content after making OTA contaminated feed, OTA contents was 4 ppm in feeds.

  1. I am not sure did I get it right, but if you mention fertilized eggs, it means that you had some roosters in the experimental flock? You are mentioning only female broiler breeders?

Answer: Thank you for pointing this out. Broiler breeders are already artificially inseminated. After that, OTA-contaminated feed was fed to compare the differences of fertility and hatchability.

  1. Specific comments

I marked revised part with yellow block and deleted part with red block

L 16 : penicillium with capital letter (as in L 53)

Answer: Thank you for pointing this out. I have revised this part.

L 34-35 : birds received up to 1ml intramuscularly? as in 2.3.; it was in total around 4ml?

Answer: Thank you for pointing this out. I intramuscularly injected 1ml per 1kg of body weight. Therefore, the same amount was not injected because each body weight was different. Also nowhere did I mention that I injected 4 ml. You may have misunderstood OTA 4 mg.

L 43 : sentence needs revision (is numerically improved parameters?)

Answer: Thank you for pointing this out. I have received this part.

L 79 : no need for - behind LPS

Answer: Thank you for pointing this out. As reviewers say, this part may not be necessary. However, since LPS was co-injected to maximize the effect of OTA, I feel I have to mention this part in the introduction. In addition, in the previous paper, LPS was mentioned in the introduction in the paper in which LPS was co-injected to increase the mycotoxin effect (Xu, Eicher and Applegate, 2011). Therefore, it would be better to mention it as it is now in the introduction, and thus please consider it again.

Xu, L., Eicher, S. D., & Applegate, T. J. (2011). Effects of increasing dietary concentrations of corn naturally contaminated with deoxynivalenol on broiler and turkey poult performance and response to lipopolysaccharide. Poultry Science, 90(12), 2766-2774.

L 98, 101 : ittalic Aspergillus

Answer: Thank you for pointing this out. I have received this part.

Last sentence in 2.3. - "it was fixed to..." needs revision

Answer: Thank you for pointing this out. I have received this part.

L 111 : shell less?

Answer: Thank you for pointing this out. Shell less eggs mean eggs with no egg shell like below picture.

The part you pointed out is one of the abnormal eggs.

L 115-116 : ...small eggs under 48g eggs?

Answer: Thank you for pointing this out. The part you pointed out is one of the abnormal eggs. We counted normal eggs for fertility and hatchability excluding abnormal eggs.

L 117 : what is a breeding room?

Answer: Thank you for pointing this out. I removed the parts you pointed out

L 134 : here mentioned formula is related to?

Answer: Thank you for pointing this out. There is no formula. As shown in the photo below, the manufacturer pre-divided the steps and we compared the egg yolk color according to this standard.

L 144 : slaughtered broiler breeders - males? females?

Answer: Thank you for pointing this out. That animals were all female used in this study. We didn’t use male broiler breeders.

L 150 : ahsed?

Answer: Thank you for pointing this out. I have revised it

L 157 : how was the live weight of the bird determined throughout the experiment?

Answer: Thank you for pointing this out. To determine organ weight, chickens removed from cages are live-weighed and then sacrificed and organs are weighed separately.

L 164 : which subdermal vein?

Answer: Thank you for pointing this out. I have changed this part from subdermal to wing. We collected blood from their wing vein

L 163-169 : please mention first full name and then abbreviation in the brackets

Answer: Thank you for pointing this out. I have changed this part

L 173 : how were the samples grounded?

Answer: Thank you for pointing this out. I have changed this part

L 195, 211,225,236,249,262 : please change the sentences "By supplementation")

Answer: Thank you for pointing this out. I have changed this part

Tables 2-5, what does N=10 means?

Answer: Thank you for pointing this out. That means replicate

L 262 : OTA in the feed? accumulation not mentioned before in the Material and Methods?

Answer: Thank you for pointing this out. I added some information about feed to material and methods

L 277 : dot

Answer: Thank you for pointing this out. I have removed that dot

L 311-312 : sentence needs revision

Answer: Thank you for pointing this out. I have changed this part

L 344 : cholestrerol?

Answer: Thank you for pointing this out. I have changed this part

Reviewer 2 Report

All comments are in the attached file.

It is important that authors include mode action in discussion and improve the conclusion

Author Response

Reviewer #2:
I marked revised part with yellow block and deleted part with red block

  1. Would be interesting perform also Dunnett test with only positive control

Answer: Thank you for pointing this out. I agree with your interesting suggestions. But we also wanted to see the difference between them by using two levels, two different additives, so we used the rigorous statistic Tukey. Please consider it again.

  1. Exclude T3 and T4

Answer: Thank you for pointing this out. I have revised it

  1. Was there a reduction in feed consumption? It can help explain the lower egg production too.

Answer: Thank you for pointing this out. I agree with your interesting suggestions. Actually, chickens exposed to OTA feed seem to have lower feed intake in this study. Likewise, pervious studies showed that mycotoxin decreased feed intake. However, we didn’t measure feed intake in this study and thus didn’t have any data about feed intake.  

  1. I missed the action mechanism of the toxin binders used in this study. Authors could explain the mechanism here to clarify how these substances can improve the egg production.

Answer: Thank you for pointing this out. I have added mechanism of toxin binder in this parts.

  1. authors could explain better here the role of plant extracts https://www.ncbi.nlm.nih.gov/pmc/articles/PMC6899290/

Answer: Thank you for pointing this out. I have added role of plant extract in this parts.

  1. resume the conclusions. It seems more a paragraph from discussion item.

Answer: Thank you for pointing this out. I have revised this parts

Reviewer 3 Report

 In this study, ochratoxin A and LPS-challenged broiler breeders were used to test the effects of toxin binders (such as clay minerals and charcoals) on laying performance, egg quality, relative organ weight, bone mineralization, blood traits, and toxin concentration. Although the study is interesting, a few suggestions and concerns must be addressed before it can be accepted.

1.      The introduction is insufficient; the authors need to have given a justification for their decision to combine feed additives in these trials and how they determined the current dose of OTA, LPS, or both.

2.      The method used to choose the dose of every compound is not disclosed. This need clarification

3.      There is obviously a need for improvement in the introduction.

4.      Sorry, but the justifications presented in the introduction do not support the aim of the study.

5.      Materials and methods: More information is required on the preparation and addition of each product by the authors to the diets, which served as the subjects' physical form during the entire experiment.

6.      OTA free basal diet (Negative control), How could you guarantee that this diet is OTA-free?  Did you examine the OTA in the diet within control?

7.       Feed additive 1 (FA1) was composed of montmorillonite, clinoptinolite, and charcoal. Additional details regarding the percentage of every substance are needed.

8.      …….and 1 mL of E.coli O55:B5 LPS per kg of body weight was intramuscularly injected. Why did you just administer this treatment using E. coil?

9.      How did you mix the additive and compounds into the diets? What is physical feed form eg. Mash or pelleted ..of diets which offered to the birds?

10.   Have you examined the levels of ochratoxin A in each diet? analysis is required for all diets

11. Results and discussion : What were the mechanisms of action that affected liver function, egg quality, and bone mineralization? The speculated causes that can potentially lead to these alterations weren't explained.   

Author Response

Reviewer #3:
I marked revised part with yellow block and deleted part with red block

  1. The introduction is insufficient; the authors need to have given a justification for their decision to combine feed additives in these trials and how they determined the current dose of OTA, LPS, or both.

Answer: Thank you for pointing this out. I have revised it. Moreover, each challenge dose was proceeded by referring to the level at which the negative effect of the challenge appeared in the previous paper. These were described in Materials and Methods, respectively.

  1. The method used to choose the dose of every compound is not disclosed. This need clarification

Answer: Thank you for pointing this out. The level of additives in animal science is 0.1%. This level does not adversely affect economics in the farm and is a very common level that can be effective to animals’ health. Therefore, based on this criterion, the level desired by the company was compared with each other.

  1. There is obviously a need for improvement in the introduction.

Answer: Thank you for pointing this out. I have revised it.

  1. Sorry, but the justifications presented in the introduction do not support the aim of the study.

Answer: Thank you for pointing this out. I have revised it.

  1. Materials and methods: More information is required on the preparation and addition of each product by the authors to the diets, which served as the subjects' physical form during the entire experiment.

Answer: Thank you for pointing this out. I have added information of FA1 and FA2 in material and methods.

  1. OTA free basal diet (Negative control), How could you guarantee that this diet is OTA-free? Did you examine the OTA in the diet within control?

Answer: Thank you for pointing this out. I showed the OTA levels in feed to table 9.

  1. Feed additive 1 (FA1) was composed of montmorillonite, clinoptinolite, and charcoal. Additional details regarding the percentage of every substance are needed.

Answer: Thank you for pointing this out. I have added information of FA1 and FA2 in material and methods.

  1. …….and 1 mL of E.coli O55:B5 LPS per kg of body weight was intramuscularly injected. Why did you just administer this treatment using E. coil?

Answer: Thank you for pointing this out. In previous study, they used E.coli LPS to maximize negative effect of mycotoxin and actually found more synergetic effect (Xu, Eicher and Applegate, 2011). Therefore, we also used E.coli LPS to maximize negative effects of OTA as above mentioned paper and estimated positive effects of feed additives we used in this study.

  1. How did you mix the additive and compounds into the diets? What is physical feed form eg. Mash or pelleted ..of diets which offered to the birds?

Answer: Thank you for pointing this out. Form of almost feed additives is powder. Therefore, we made a mash basal diet and then added feed addtives to mash basal diet.

  1. Have you examined the levels of ochratoxin A in each diet? analysis is required for all diets

Answer: Thank you for pointing this out. I showed the OTA levels in feed to table 9.

  1. Results and discussion : What were the mechanisms of action that affected liver function, egg quality, and bone mineralization? The speculated causes that can potentially lead to these alterations weren't explained.

Answer: Thank you for pointing this out. I have revised these part to clarity the effects of feed additives.

Round 2

Reviewer 1 Report

Dear authors,

thank you for taking into consideration all the comments given by the reviewer. The only issue still remaining is: it seems to me that there was a misunderstanding between you and the reviewer related to  L 195, 203, 211, 225, 236, 249 and 262. Those sentences were mentioned by the reviewer because all were the almost the same, starting with same words; but also after your review, they are still the same (starting with same words). I suggest you write them in such a way that they don't all start with "By supplementation of natural feed additives (FA1 and FA2)...).

Dear Editor,

the authors took into consideration all the comments given by the reviewer, there is still one language issue that could be addressed by the authors.

After this change, the manuscript can be accepted.

Author Response

Reviewer #1:

1. thank you for taking into consideration all the comments given by the reviewer. The only issue still remaining is: it seems to me that there was a misunderstanding between you and the reviewer related to  L 195, 203, 211, 225, 236, 249 and 262.

Those sentences were mentioned by the reviewer because all were the almost the same, starting with same words; but also after your review, they are still the same (starting with same words).

I suggest you write them in such a way that they don't all start with "By supplementation of natural feed additives (FA1 and FA2)...).

Answer: I have delete the sentence start with “by supplementation of natural feed additives” and I directly jump into results
